# Development of a UHPLC-MS/MS Method for Quantitative Analysis of Aflatoxin B_1_ in *Scutellaria baicalensis*

**DOI:** 10.3390/toxins17090473

**Published:** 2025-09-21

**Authors:** Yuanfang Liu, Cuiping Zeng, Ying-Ying Li, Jiayu Guo, Jinming Xu

**Affiliations:** 1Department of Chemistry, Zhengzhou Normal University, No. 6, Yingcai Street, Huiji District, Zhengzhou 450044, China; liyingying@zznu.edu.cn (Y.-Y.L.); guojiayu921@163.com (J.G.); huanysza@163.com (J.X.); 2College of Chemistry, and Pingyuan Laboratory, Zhengzhou University, Zhengzhou 450001, China; zcp13461653017@163.com

**Keywords:** aflatoxin B_1_, *Scutellaria baicalensis*, UHPLC-MS/MS, quantitative analysis, method validation

## Abstract

Aflatoxin B_1_ (AFB_1_) contamination in *Scutellaria baicalensis* poses a serious threat to the safety of traditional Chinese medicinal products. In this study, a sensitive and reliable ultra-high performance liquid chromatography-tandem mass spectrometry (UHPLC-MS/MS) method was developed for the quantitative determination of AFB_1_ in *Scutellaria baicalensis*. Method optimization included selection of chromatographic columns, mobile phase composition, and mass spectrometric parameters. Sample pretreatment was also optimized to reduce matrix interference and enhance extraction efficiency. The method showed excellent linearity (R^2^ > 0.999) in the range of 0.1–10.0 µg/L, with a limit of detection (LOD) of 0.03 µg/kg and a limit of quantification (LOQ) of 0.10 µg/kg. Precision and recovery studies demonstrated good repeatability and accuracy, with intra- and inter-day relative standard deviations (RSDs) below 5.2% and recoveries ranging from 88.7% to 103.4%. Application of the method to six commercial *Scutellaria baicalensis* samples revealed detectable AFB_1_ in two samples, though all levels were below national safety limits. This method provides a robust tool for routine monitoring of AFB_1_ in herbal medicines and supports the establishment of quality control systems for *Scutellaria baicalensis*.

## 1. Introduction

*Scutellaria baicalensis* (commonly known as Chinese skullcap) is a traditional medicinal herb widely used in East Asia, especially in Chinese medicine, for its anti-inflammatory, antioxidant, and hepatoprotective properties. Its dried root is the main medicinal part and contains abundant flavonoids such as baicalin and wogonoside. Aflatoxin B_1_ (AFB_1_), produced primarily by *Aspergillus flavus* and *Aspergillus parasiticus*, is a highly toxic Group I carcinogen (Figure 1). It frequently contaminates agricultural products and traditional Chinese medicinal herbs, posing serious health risks such as hepatotoxicity, immunosuppression, and genotoxicity [1]. Under hot and humid storage conditions, traditional Chinese medicinal herbs—especially starchy, oily seeds, rhizomes, and wild-collected species—are prone to fungal growth and subsequent mycotoxin production. The contamination level of AFB_1_ is significantly influenced by region and storage environment; for instance, the detection rate in southern provinces for *Scutellaria baicalensis* can reach 10–30% [2,3]. Traditional drying techniques often fail to completely deactivate fungi and effectively suppress toxin production [4,5].

To address the demand for trace-level AFB_1_ detection in complex matrices such as *Scutellaria baicalensis*, this study aims to establish a highly sensitive and specific quantitative method based on ultra-high performance liquid chromatography-tandem mass spectrometry (UHPLC-MS/MS). This method seeks to overcome the limitations of traditional HPLC approaches, such as insufficient sensitivity and weak anti-interference capability, while optimizing the sample pretreatment procedures to reduce matrix effects and enhance recovery. This provides reliable technical support for fungal toxin monitoring in herbal medicine.

Due to susceptibility to moisture, *Scutellaria baicalensis* is highly prone to mildew and subsequent accumulation of AFB_1_ during cultivation, harvesting, and storage. Although AFB_1_ detection technology has evolved, achieving the right balance between sensitivity, matrix interference elimination, and cost-efficiency remains a critical challenge. Stringent international regulatory limits (e.g., 2 µg/kg for herbal materials in the European Union) [6] also demand improved analytical solutions. Table 1 summarizes the current mainstream detection methods for Aflatoxin B1, each with its own limitations. Enzyme-Linked Immunosorbent Assay (ELISA), while simple, suffers from high matrix interference and false positives [7]; HPLC with fluorescence detection (HPLC-FLD) lacks sufficient sensitivity (LOD often 1–5 µg/kg) and requires complex derivatization [8,9]; immunoaffinity column-fluorescence methods offer good sensitivity (LOD 0.1–0.5 µg/kg) but at high cost; novel techniques like surface-enhanced Raman scattering (SERS) and electrochemical sensors show promise but face challenges of reproducibility and stability; and GC-MS is highly sensitive but entails complex derivatization and high cost.

UHPLC-MS/MS, with advantages of high resolution, sensitivity, and multi-reaction monitoring (MRM) capability, has emerged as a mainstream tool for mycotoxin analysis [10]. Nevertheless, current methods applied to herbal matrices such as *Scutellaria baicalensis* still suffer from cumbersome pretreatment, significant matrix effects, and signal suppression of trace-level analytes. Therefore, method optimization is imperative to improve analytical performance and practical applicability. Although several validated methods have been established for AFB_1_ detection in cereals and food matrices, they are not directly applicable to *Scutellaria baicalensis* due to its unique and complex matrix composition. The presence of abundant flavonoids and polysaccharides in this herb poses significant matrix effects, such as ion suppression and co-elution, which can compromise the accuracy and sensitivity of standard protocols. Therefore, a dedicated method tailored for *Scutellaria baicalensis* is essential.

This study develops a UHPLC-MS/MS method specifically tailored for quantitative detection of trace AFB_1_ in *Scutellaria baicalensis*. Optimization includes chromatographic separation, mass spectrometric parameters (mobile phase, elution program, MRM ion pairs), and sample pretreatment steps (extraction solvent, extraction time, solid–liquid ratio). The method is validated for linearity, LOD, LOQ, precision, and recovery, and applied to the assessment of AFB_1_ contamination in *Scutellaria baicalensis* samples from different regions. This work provides methodological reference for mycotoxin analysis in herbal medicine and supports the establishment of quality control systems for *Scutellaria baicalensis*.

## 2. Results and Discussion

### 2.1. Optimization of UHPLC-MS/MS Parameters

#### 2.1.1. Ionization Mode Selection and Collision Energy Optimization

To improve AFB_1_ detection sensitivity and accuracy, various chromatographic columns, mobile phase compositions, and ion transitions were evaluated. The Agilent ZORBAX Eclipse Plus C18 column demonstrated optimal peak shape and resolution [11]. The addition of 0.1% formic acid to both aqueous and organic phases significantly enhanced ionization efficiency. Methanol was chosen over acetonitrile due to improved peak intensity and stability. The optimal ion transition for quantification was *m*/*z* 313.2 →241.2, with *m/z* 313.2→285.1 used for confirmation. These transitions provided strong signal intensity and selectivity under positive ESI mode. As shown in Figure 2, the ion abundance varies with different collision energies. The highest ion abundance was observed at 120 V, which was consequently selected as the optimal collision energy.

Through systematic optimization of collision energy (105–135 V), the highest abundance of the target ion was observed at 120 V (Table 2), which was consequently selected as the optimal collision energy.

#### 2.1.2. Optimization of Collision Energy

Collision energy (CE) is a critical parameter affecting ion fragmentation efficiency. Proper optimization of CE enables more effective dissociation of precursor ions into target product ions, significantly enhancing their response intensity. For AFB_1_ analysis, the CE optimization range was set at 24–32 eV with 2 eV increments to systematically evaluate the abundance variations in product ions under different energy conditions. Experimental results demonstrated that: The quantitative ion transition (*m*/*z* 313.2→285.1) reached maximum abundance at 24 eV; the qualitative ion transition (*m*/*z* 313.2→241.1) achieved peak intensity at 32 eV. As illustrated in Figure 3 and Figure 4, the abundance profiles of product ions *m*/*z* 285.1 and *m*/*z* 241.1 are displayed corresponding to different CE values.

#### 2.1.3. Selection of Precursor and Product Ions

Following collision energy optimization, the protonated molecule [M+H]^+^ of AFB_1_ (*m*/*z* 313.2) was selected as the precursor ion. Through collision-induced dissociation (CID) and subsequent product ion scanning, fragment ions with high response intensity and excellent stability were identified. The final transitions were established as: Quantitative transition: *m*/*z* 313.2→285.1. Qualitative transition: *m*/*z* 313.2→241.1(Figure 5). This selective ion pairing strategy significantly enhances both detection specificity (reduced matrix interference) and analytical sensitivity (improved S/N ratio).

#### 2.1.4. MRM Parameter Configuration

Through systematic optimization, the protonated molecular ion [M+H]^+^ (*m*/*z* 313.2) was selected as the precursor ion, demonstrating the strongest response in positive ion mode. Maximum ion abundance was achieved at a fragmentor voltage of 120 V. The quantitative transition (313.2→285.1) reached peak abundance at 24 eV collision energy, while the qualitative transition (313.2→241.1) peaked at 32 eV (Figure 3 and Figure 4), with complete parameters detailed in Table 3. Additional product ion channels (including *m*/*z* 285.1 and 241.1) were evaluated for response intensity, selectivity, and stability to identify optimal MRM combinations (Figure 6).

### 2.2. Optimization and Evaluation of Extraction Conditions

#### 2.2.1. Effect of Stirring Time on Extraction Efficiency

As shown in Figure 7, the extraction yield of AFB_1_ exhibited an initial increase followed by stabilization and subsequent decline with prolonged stirring time: At 20 min, the concentration was 26.45 μg·kg^−1^, which significantly increased to 28.89 μg·kg^−1^ by 30 min (*p* < 0.05). No statistically significant difference was observed between 30 min (28.89 μg·kg^−1^) and 40 min (28.85 μg·kg^−1^) (*p* > 0.05), indicating a stabilization phase. However, further extension to 50 min reduced the yield to 25.60 μg·kg^−1^, suggesting potential co-extraction of interfering compounds or analyte degradation.

Therefore, 30 min was selected as the optimal stirring duration, a finding consistent with the “increase-peak-decline” pattern reported by Zhao et al. [12] in their response surface methodology optimization of aflatoxin extraction from peanuts for rapid detection applications.

#### 2.2.2. Optimization of Extraction Solvent Composition

Under fixed stirring duration (30 min), the solvent composition was systematically evaluated, with pure acetonitrile demonstrating optimal extraction efficiency (28.55 μg/kg AFB_1_). A significant negative correlation was observed as increasing water content reduced yields to 10.50 μg/kg (4% water) and 4.20 μg/kg (16% water), leading to selection of pure acetonitrile, consistent with Li et al. [13] in ELISA-based AFB_1_ detection from soy sauce. The results are presented in Figure 8, showing the clear inverse relationship between water proportion and extraction efficiency.

#### 2.2.3. Optimization of Solid-to-Liquid Ratio

With fixed magnetic stirring for 30 min and pure acetonitrile as the extraction solvent, the results (Figure 9) demonstrated: AFB_1_ concentrations remained stable at 20–25 μg/kg with ratios of 1:5 to 1:7 (g/mL), but increased significantly to 56.72 μg/kg at 1:8 and peaked at 67.86 μg/kg with a 1:9 ratio. This indicates that smaller solid-to-liquid ratios (1:9) provide more sufficient solvent volume to enhance AFB1 release, a finding consistent with Li et al. [13] in their ELISA-based detection of aflatoxin B_1_ in soy sauce. Therefore, the 1:9 ratio was selected as optimal.

### 2.3. Method Validation Results

The calibration curve showed excellent linearity in the 0.05-50.00 µg/kg range, with R^2^ > 0.999 (Figure 10). The LOD and LOQ of AFB_1_ in *Scutellaria baicalensis* were 0.03 µg/kg and 0.10 µg/kg, respectively, which meet regulatory requirements for trace detection (Table 4). Precision analysis revealed intra-day RSDs of 2.1–4.5% and inter-day RSDs of 3.0–5.2%, indicating good repeatability. Average recoveries ranged from 88.7% to 103.4% at three spiking levels (0.5, 2.0, and 5.0 µg/kg), with RSDs below 6.0%, demonstrating acceptable accuracy and reliability of the method.

Spiked recovery tests were conducted by adding AFB_1_ standard solutions at three concentration levels (0.10 μg/kg, 2.00 μg/kg, and 10.00 μg/kg) to pre-processed *Scutellaria baicalensis* samples. Under the optimized conditions, the method demonstrated excellent accuracy with a mean recovery rate of 96.10% and high precision (RSD = 1.9%, n = 6). These results confirm the reliability and accuracy of the established method for quantitative determination of AFB_1_ in *Scutellaria baicalensis* (Table 5).

### 2.4. AFB_1_ Contamination in Commercial Samples

A total of 12 batches of *Scutellaria baicalensis* from different provinces were analyzed, including samples from Lanzhou (Gansu), Bozhou (Anhui), Yancheng (Jiangsu), Anguo (Hebei), and Chengdu (Sichuan). Among them, 3 batches of samples were detected with AFB_1_, accounting for 25% of the total. The concentration of AFB_1_ in Lanzhou (Gansu) was the highest at 6.28 μg/kg, which was 1.25 times higher than the limit specified in the Chinese Pharmacopoeia. The AFB_1_ content in samples from other regions did not exceed the limit. The observed regional differences in AFB_1_ contamination may be associated with several factors. Firstly, cultivation conditions such as soil type and irrigation practices can influence fungal growth. Secondly, post-harvest storage conditions, including temperature and humidity control, play a critical role in preventing mycotoxin accumulation. Finally, climatic variations among regions (e.g., relative humidity and average temperature during harvest and storage) may also contribute to the observed discrepancies. Further studies with larger sample sizes and detailed metadata collection will help to clarify these associations. As shown in Table 6.

## 3. Conclusions

A sensitive and reliable UHPLC-MS/MS method was successfully developed for the quantitative analysis of AFB_1_ in *Scutellaria baicalensis*. The method features low detection limits, high precision, and good recovery. It is suitable for routine monitoring of AFB_1_ contamination in herbal medicine. This method can aid in establishing robust quality control systems for *Scutellaria baicalensis* and contributes to ensuring the safety of traditional Chinese medicinal products. Despite the method’s strong performance, several limitations remain. The high cost of UHPLC-MS/MS instrumentation may limit its widespread adoption in resource-limited settings. Furthermore, the method’s robustness across different laboratories and operators has not yet been fully validated. Future research should focus on inter-laboratory validation, long-term stability testing, and the development of simplified sample preparation protocols. Large-scale surveillance of *Scutellaria baicalensis* from broader geographic sources will also be essential to ensure quality and safety consistency.

## 4. Materials and Methods

### 4.1. Materials and Reagents

Acetonitrile and methanol (MS grade) were obtained from Thermo Fisher Scientific (Waltham, MA, USA); analytical-grade acetonitrile was purchased from Tianjin Kermel Chemical Reagent Co., Ltd. (Tianjin, China). Purified water was sourced from Watsons Beverage Technology Co., Ltd. (Guangzhou, China). The Aflatoxin B_1_ standard solution (purity 98%) was obtained from Hebei Bailin Superfine Materials Co., Ltd. (Langfang, China) Six *Scutellaria baicalensis* decoction pieces were sourced from different provinces as detailed in Table 7.

### 4.2. Instruments and Equipment

The instrumentation included a QW-DCY-KS12A nitrogen evaporator (Hangzhou Qiwei Instruments Co., Ltd., Hangzhou, China), EL204-IC electronic balance (Shanghai Yueping Scientific Instrument Co., Ltd., Shanghai, China), G6470B UHPLC-MS/MS system (Agilent Technologies, Santa Clara, CA, USA), 150 high-speed pulverizer (Yongkang Minye Industry & Trade Co., Ltd., Yongkang, China ), H1850 centrifuge (Xiangyi Laboratory Instrument Development Co., Ltd., Changsha, China)), 906-201 vortex mixer (Hefei Aibensen Scientific Instruments Co., Ltd., Hefei, China), and 101-00BS electric thermostatic drying oven (Shanghai Lichen Bangxi Instrument Technology Co., Ltd., Shanghai, China).

### 4.3. Preparation of Standard Solutions and Sample Pretreatment

The AFB_1_ standard stock solution was prepared by diluting the reference standard to a concentration of 1.0 mg/L with methanol and stored in amber glass bottles at 4 °C. A series of working standard solutions at concentrations of 0.1, 0.5, 1.0, 2.0, 5.0, and 10.0 µg/kg [14,15,16,17] were obtained by serial dilution.

Each *Scutellaria baicalensis* sample was pulverized, passed through a 40-mesh sieve, and stored in a sealed, light-protected container. Accurately weighed 2.000 g of sample was transferred into a 50 mL centrifuge tube. Then, 20 mL of 80% acetonitrile aqueous solution was added, and the tube was vortexed for 3 min. Ultrasonic extraction was performed at room temperature for 40 min, followed by centrifugation at 4500 rpm for 5 min [18]. An aliquot of 10 mL supernatant was transferred to a new centrifuge tube and evaporated to dryness under a gentle nitrogen stream at 40 °C. The residue was reconstituted with 1.0 mL methanol-water (50:50, *v*/*v*), vortexed for 1 min, and filtered through a 0.22 µm membrane filter before UHPLC-MS/MS analysis [19,20] (Figure 11).

### 4.4. Chromatographic and Mass Spectrometric Conditions

Chromatographic separation was carried out on an Agilent ZORBAX Eclipse Plus C18 column (2.1 mm × 100 mm, 1.8 µm) using gradient elution. The mobile phase consisted of water with 0.1% formic acid (A) and methanol with 0.1% formic acid (B), delivered at a flow rate of 0.3 mL/min. The elution program was as follows [21]: 0–1.5 min, linear gradient to 95% B; 1.5–4.2 min, held at 95% B; 4.2–4.3 min, returned to 25% B and re-equilibrated. As shown in Table 8.

The mass spectrometer operated in positive electrospray ionization (ESI+) mode with multiple reaction monitoring (MRM) [22]. The ion source conditions were: gas temperature 350 °C, gas flow 10 L/min, nebulizer pressure 40 psi, capillary voltage 4000 V. The precursor ion of AFB_1_ was *m*/*z* 313.2, and product ions for quantification and confirmation were *m*/*z* 241.2 and *m*/*z* 285.1, respectively. As shown in Table 9.

### 4.5. Single-Factor Experiments

1.Magnetic Stirring Time

The ratio of acetonitrile-water solution was fixed at 100:0 (*v*/*v*), and the solid-to-liquid ratio was set at 1:5 (g/mL). The stirring time gradient was set at 20, 30, 40, 50, and 60 min. Each group was tested in triplicate, and the AFB_1_ concentration was calculated based on peak area measurements using UHPLC-MS/MS.

2.Water Ratio in Acetonitrile-Water Solution

The solid-to-liquid ratio was fixed at 1:5 (g/mL), and the stirring time was set at 30 min. The acetonitrile-water solution gradient was set at 100:0, 96:4, 92:8, 88:12, and 84:16 (*v*/*v*). Each group was tested in triplicate, and the AFB_1_ concentration was determined via UHPLC-MS/MS peak area analysis.

3.Solid-to-Liquid Ratio

The acetonitrile-water solution ratio was fixed at 100:0 (*v*/*v*), and the stirring time was set at 30 min. The solid-to-liquid ratio gradient was set at 1:5, 1:6, 1:7, 1:8, and 1:9 (g/mL). Each group was tested in triplicate, and the AFB_1_ concentration was calculated based on UHPLC-MS/MS peak area measurements.

### 4.6. Method Validation

The method was validated for linearity [23], limit of detection (LOD), limit of quantification (LOQ), precision, and recovery. Linearity was assessed using matrix-matched calibration curves constructed by spiking blank *Scutellaria baicalensis* extracts with AFB_1_ at six concentration levels (0.1–10.0 µg/L), with the coefficient of determination (R^2^) required to be ≥0.999. The LOD and LOQ were determined based on signal-to-noise ratios (S/N) of 3 and 10, respectively.

Intra-day and inter-day precision were evaluated by analyzing spiked samples at three concentration levels (0.5, 2.0, 5.0 µg/kg) in six replicates, expressed as relative standard deviation (RSD%) [24]. Recovery experiments were conducted by spiking blank samples at the same levels and calculating the mean percentage of AFB_1_ recovered.

## Figures and Tables

**Figure 1 toxins-17-00473-f001:**
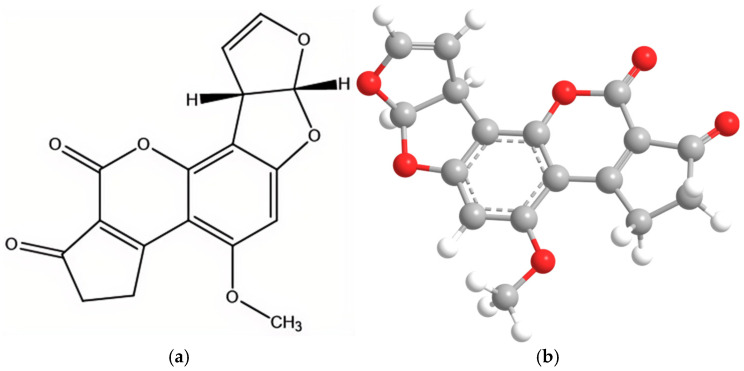
(**a**) Structural formula of AFB_1_. (**b**) Schematic diagram of the 3D structure of AFB_1_.

**Figure 2 toxins-17-00473-f002:**
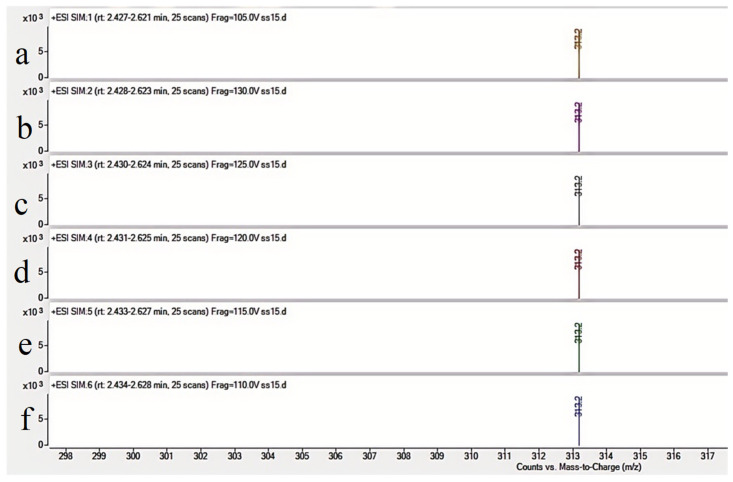
Precursor ion (*m*/*z* 313.2) abundance vs. collision energy ((**a**): 105 V (**b**): 130 V (**c**): 125 V (**d**): 120 V (**e**): 115 V (**f**): 110 V).

**Figure 3 toxins-17-00473-f003:**
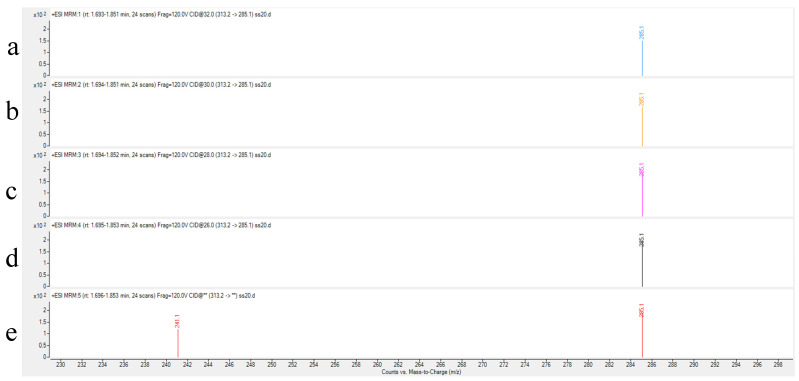
Abundance of product ion (*m*/*z* 285.1) at different collision energies. ((**a**): 32 eV (**b**): 30 eV (**c**): 28 eV (**d**): 26 eV (**e**): 24 eV).

**Figure 4 toxins-17-00473-f004:**
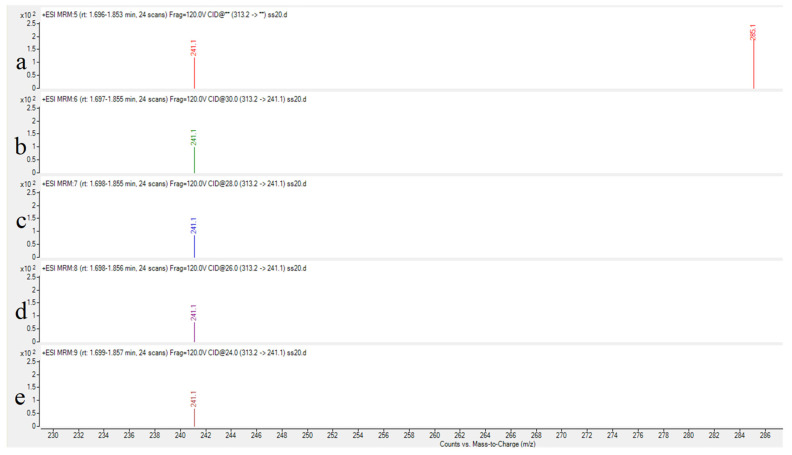
Abundance of product ion (*m*/*z* 241.1) at varied collision energies. ((**a**): 32 eV (**b**): 30 eV (**c**): 28 eV (**d**): 26 eV (**e**): 24 eV).

**Figure 5 toxins-17-00473-f005:**
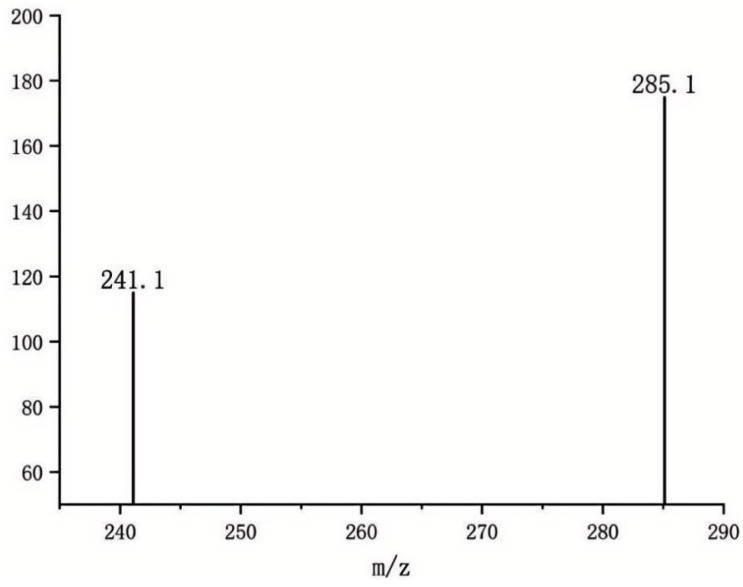
Abundance of Two Product Ions.

**Figure 6 toxins-17-00473-f006:**
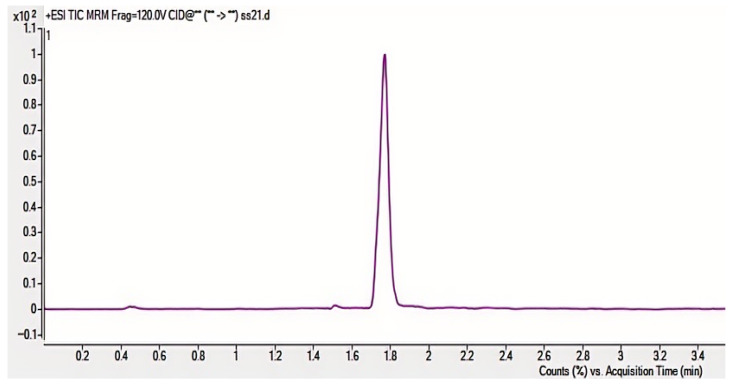
MRM ion chromatograms.

**Figure 7 toxins-17-00473-f007:**
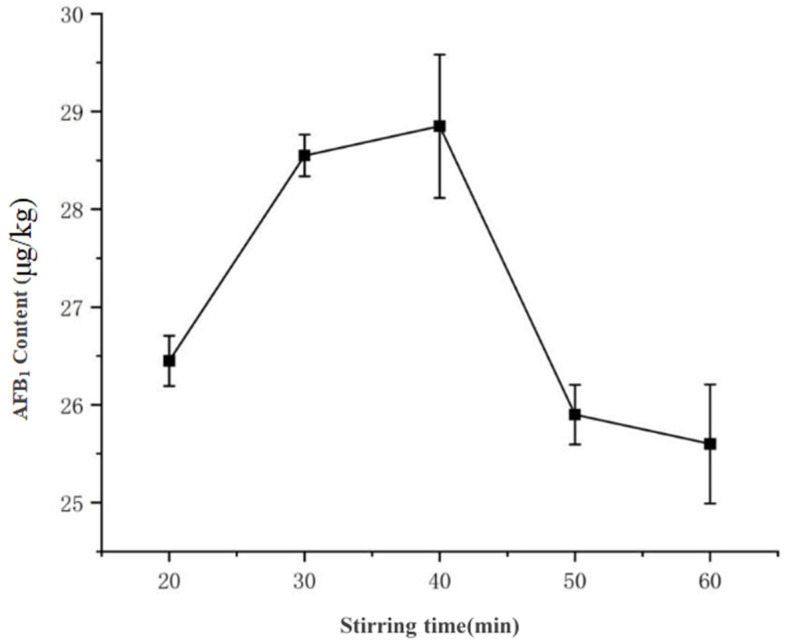
Effect of magnetic stirring duration on AFB_1_ extraction efficiency.

**Figure 8 toxins-17-00473-f008:**
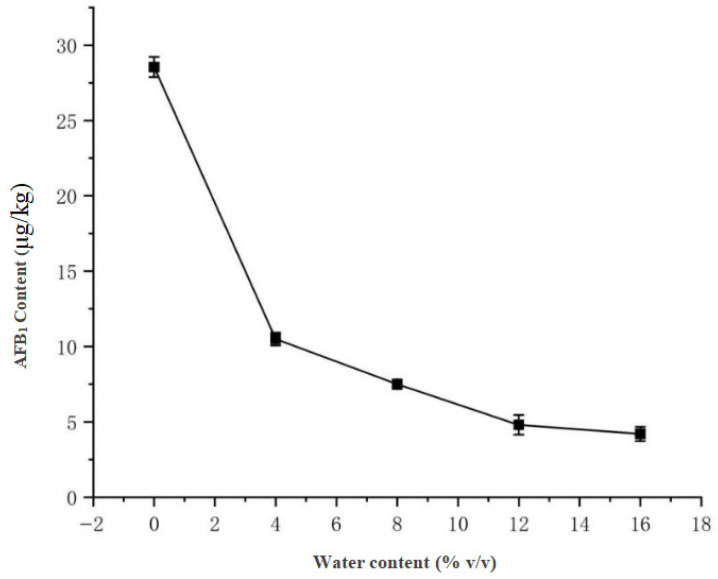
Effect of water content on AFB_1_ extraction efficiency.

**Figure 9 toxins-17-00473-f009:**
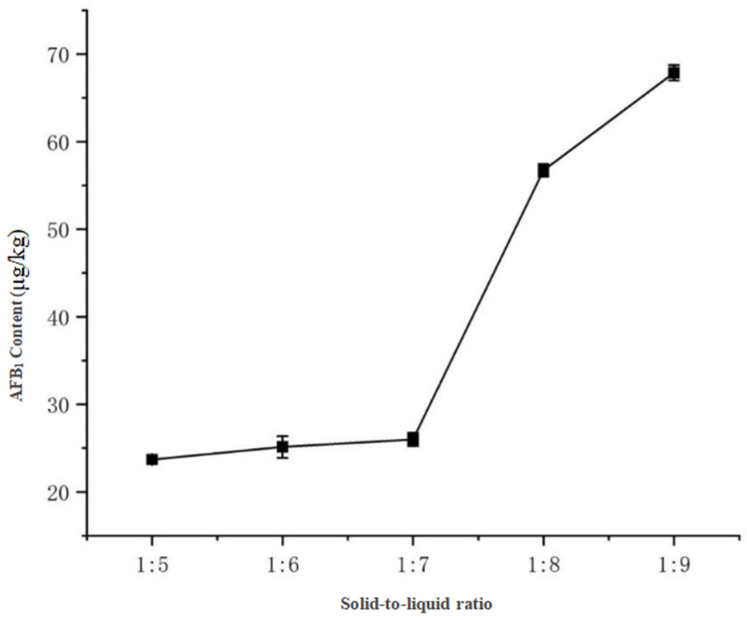
Effect of solid-to-liquid ratio on AFB_1_ extraction efficiency.

**Figure 10 toxins-17-00473-f010:**
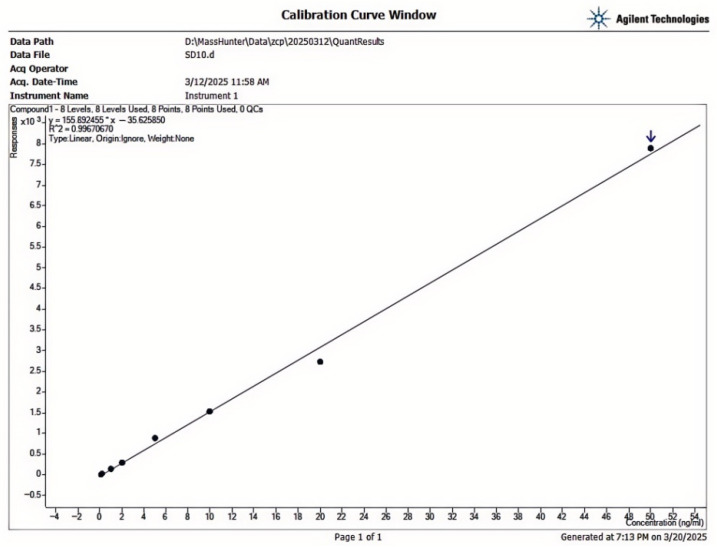
AFB_1_ Standard Curve.

**Figure 11 toxins-17-00473-f011:**
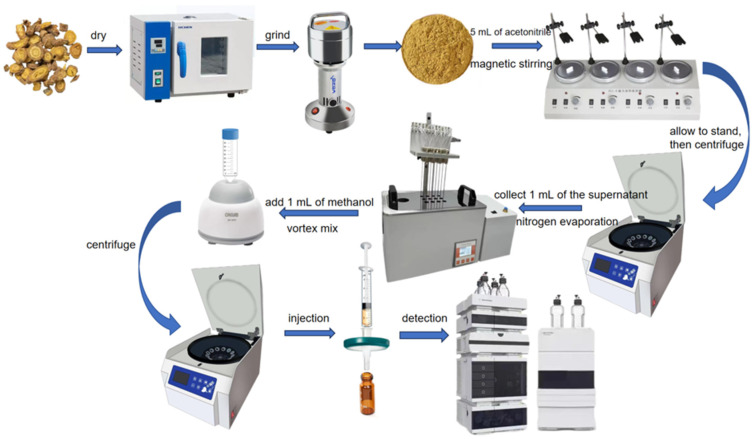
AFB_1_ Extraction Flowchart.

**Table 1 toxins-17-00473-t001:** Comparison of AFB_1_ Detection Methods in Traditional Chinese Medicines [8,9,10].

Parameter	LOD	LOQ	Precision (RSD)	Recovery Rate	Operational Complexity
ELISA	0.10–0.50 μg/kg	0.50–2.00 μg/kg	8.0–15%	60–100%	Moderate
TLC	1.00–5.00 μg/kg	5.00–10.00 μg/kg	15–25%	60–110%	Low
SERS	0.01–0.10 μg/kg	0.05–5.00 μg/kg	10–18%	75–110%	High
Electrochemical Sensor	0.005–0.051 μg/kg	0.02–0.20 μg/kg	12–20%	70–120%	Medium
GC-MS	0.02–0.10 μg/kg	0.05–0.50 μg/kg	5.0–10%	80–110%	High

**Table 2 toxins-17-00473-t002:** Abundance values corresponding to different collision energies.

Collision Energy (V)	Ion Abundance
105	40,786.76
110	41,467.34
115	41,769.26
120	43,745.19
125	8130.21
130	8234.21
135	8596.15

**Table 3 toxins-17-00473-t003:** Parameters for Multiple Reaction Monitoring (MRM) Mode.

Analyte	Retention Time (min)	Ionization Form	Precursor Ion (*m*/*z*)	Quantifier Ion (*m*/*z*)	Qualifier Ion (*m*/*z*)	Fragmentor Voltage (V)	Collision Energy (eV)
AFB_1_	1.8	[M+H]^+^	313.1	285.1	241.1	**120**	24/32

**Table 4 toxins-17-00473-t004:** Validation Parameters for the Quantitative Determination of AFB_1_ in *Scutellaria baicalensis* by UHPLC-MS/MS.

	Linear Range	LOD (μg/kg)	LOQ (μg/kg)	Precision (RSD)	Accuracy
AFB_1_	0.05–50.00 μg/kg	0.01	0.03	1.9%	3.7%

**Table 5 toxins-17-00473-t005:** Spiked recovery test results of AFB_1_ in *Scutellaria baicalensis*.

Original Concentration/(μg/kg)	Sample Volume /mL	Spiked Amount /(μg/kg)	Spiked Volume /mL	Measured Value /(μg/kg)	Recovery Rate/%	Average Recovery Rate/%	(RSD)/%
14.60	1.00	0.10	1.00	14.69	94.03	96.10	1.9
14.60	1.00	2.00	1.00	16.56	97.84		
14.60	1.00	10.00	1.00	24.24	96.43		

**Table 6 toxins-17-00473-t006:** AFB_1_ levels in *Scutellaria* from different regions.

Origin	Sample Appearance	AFB_1_ Content (μg/kg)	Exceedance Multiple	Contamination Level
Ji’an, Jiangxi	Yellow	27.40	5.48	Severe contamination
Yan’an, Shanxi	Yellow	14.60	2.92	Moderate contamination
Baotou, Inner Mongolia	Yellow	24.70	4.94	Severe contamination
Taiyuan, Shanxi	Yellow-brown	8.10	1.62	Mild contamination
Lanzhou, Gansu	Light yellow	Not detected (ND)	Not detected (ND)	No contamination
Huaibei, Anhui	Yellow	20.40	4.08	Severe contamination

**Table 7 toxins-17-00473-t007:** Appearance and origin of *Scutellaria baicalensis* samples from different regions.

Sample Number	Product Appearance	Place of Origin
1	Yellow	Ji’an City, Jiangxi Province
2	Yellow	Yan’an City, Shaanxi Province
3	Yellow	Baotou City, Inner Mongolia
4	Yellow-brown	Taiyuan City, Shanxi Province
5	Light yellow	Lanzhou City, Gansu Province
6	Yellow	Huaibei City, Anhui Province

**Table 8 toxins-17-00473-t008:** Mobile Phase Elution Gradient.

Time (min)	A (%)	B (%)
0	75	25
1.5	5	95
4.2	5	95
4.3	75	25
6.0	75	25

**Table 9 toxins-17-00473-t009:** Mass Spectrometer Parameters.

Parameter Name	Parameter Value
Sheath Gas Pressure (arb)	25
Auxiliary Gas Pressure (arb)	5
Sweep Gas Pressure (arb)	0
Spray Voltage (kV)	3.00
Ion Transfer Tube Temp. (°C)	300
Auxiliary Heater Temp. (°C)	350
Scan Range (*m*/*z*)	50–800
Resolution	100,000

## Data Availability

The data presented in this study are available on request from the corresponding author. (Authors are going to explore further function and the data will be available on request).

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
