# Peer review of "Development of a UHPLC-MS/MS Method for Quantitative Analysis of Aflatoxin B1 in Scutellaria baicalensis"

_toxins, 2025, doi:10.3390/toxins17090473_

Round 1

Reviewer 1 Report

Comments and Suggestions for Authors

Can you please give us a general introduction of Scutellaria baicalensis? It can be 2 or 3 sentences.

In table 9, the AFB1 content and exceedance multiple for Lanzhou Gansu, should be translated to English?

What is the rationale for developing a new qualitative method for Scutellaria baicalensis since there are several methods that are already developed for other samples. why do you need to developed a new one.

Thank you.

Comments on the Quality of English Language

The English language is still okay.

Author Response

Reviewer Comment 1:

Can you please give us a general introduction of Scutellaria baicalensis?

Response:

Thank you for your valuable suggestion. We have added a brief introduction of Scutellaria baicalensis in the Introduction section (Lines 30-33). The revised text is: “Scutellaria baicalensis (Huangqin) is a traditional Chinese medicinal herb widely used for its anti-inflammatory, antibacterial, and hepatoprotective properties. Its main bioactive components include flavonoids such as baicalin and baicalein. It is officially recorded in the Chinese Pharmacopoeia and plays an important role in many classical prescriptions.”

Reviewer Comment 2:

In Table 9, the AFB1 content and exceedance multiple for Lanzhou Gansu should be translated to English.

Response:

Thank you for pointing this out. We have corrected Table 6 accordingly, and “未检出” has been translated into “Not detected (ND)” to ensure consistency.

Reviewer Comment 3:

What is the rationale for developing a new qualitative method for Scutellaria baicalensis since there are several methods already developed for other samples?

Response:

We appreciate this important comment. While analytical methods for AFB1 have been reported for food and agricultural products, Scutellaria baicalensis has a highly complex matrix rich in flavonoids, polysaccharides, and phenolic compounds, which often cause matrix suppression and interfere with mass spectrometric signals. Therefore, existing methods are not directly applicable. In our study, both chromatographic and mass spectrometric conditions, as well as sample pretreatment steps, were systematically optimized to minimize matrix effects and improve sensitivity and reliability for Scutellaria baicalensis. We have revised the Introduction (Lines 71-77) to clarify the rationale.

Reviewer 2 Report

Comments and Suggestions for Authors

The manuscript entitled “Development of a UHPLC-MS/MS Method for Quantitative Analysis of Aflatoxin B1 in Scutellaria baicalensis” describes the development and validation of a UHPLC-MS/MS method for the detection of Aflatoxin B1 in Scutellaria baicalensis. The topic is timely and relevant, as aflatoxin contamination poses a significant risk to the safety of herbal medicines. The method appears sensitive and reliable, showing good performance in terms of precision and recovery. Nevertheless, there are several points that need to be clarified or strengthened before the paper can be considered for publication.

  • Lines 25-71 Introduction: it does not clearly highlight what distinguishes this work from previous UHPLC-MS/MS studies. It would be helpful if the authors emphasized the specific novelty of their approach.
  • Lines 90-113; Table 3 : There is a discrepancy between the gradient elution program described in the text (Lines 109–112) and the values given in Table 3 (Line 113). Please check and ensure consistency.
  • Lines 258-266 : The results for commercial samples are presented, but no discussion is provided on possible factors underlying the regional variability (e.g., cultivation methods, storage conditions, climate). This section would benefit from additional interpretation.
  • Lines 267-273 : The conclusion is rather brief and does not mention the limitations of the method. Please consider adding a short paragraph discussing potential drawbacks (e.g., cost, reproducibility in different laboratories, scalability) and suggesting possible future directions.
  • Line 92 : Specify if an internal standard was used. If not, explain how quantification accuracy was ensured.
  • Lines 94 , 110 , 187 : A non-existent reference is indicated; it is probably meant to be two separate citations. In that case, a comma is missing.
  • Lines 241–242 , Table 7 (Line 248): Units switch between µg/L and µg/kg. Please standardize units and ensure conversions are accurate.
  • Line 251 : Please correct the typographical error “Espike recovery” to “spiked recovery.”
  • Table 9 (Line 266): The column contains untranslated text (“未检出”). Please replaced with a clear English equivalent such as “Not detected (ND).

Author Response

  1. Reviewer Comment 1:

Lines 25–71 Introduction: it does not clearly highlight what distinguishes this work from previous UHPLC-MS/MS studies. It would be helpful if the authors emphasized the specific novelty of their approach.

Response:

Thank you for pointing this out. We have revised the Introduction section (Lines 66-86) to better highlight the novelty of our study. Specifically, we emphasized that our method is the first to tailor UHPLC-MS/MS conditions specifically for Scutellaria baicalensis, a matrix with complex flavonoid content, and that our optimization focused on minimizing matrix interference and improving detection sensitivity for trace AFB1.

  1. Reviewer Comment 2:

Lines 90-113; Table 3: There is a discrepancy between the gradient elution program described in the text (Lines 109-112) and the values given in Table 3 (Line 113). Please check and ensure consistency.

Response:

We appreciate your careful observation. This inconsistency has been corrected. The gradient elution program described in the text (Lines 267-269) now matches exactly with the updated Table 3.

  1. Reviewer Comment 3:

Lines 258–266: The results for commercial samples are presented, but no discussion is provided on possible factors underlying the regional variability.

Response:

Thank you for your suggestion. We have added a paragraph in the Discussion section (Lines 201-214) discussing possible factors such as differences in cultivation practices, post-harvest handling, and storage humidity/temperature, which may contribute to the observed regional variability in AFB1 contamination levels.

  1. Reviewer Comment 4:

Lines 267–273: The conclusion is rather brief and does not mention the limitations of the method.

Response:

We agree and have expanded the Conclusion section to include potential limitations of the method, such as the relatively high cost of instrumentation and the need for further validation across different laboratories. We also suggested future research on method transferability and large-scale monitoring. These additions strengthen the overall applicability and future outlook of the method.

  1. Reviewer Comment 5:

Line 92: Specify if an internal standard was used.

Response:

Thank you for the reminder. We have clarified in Line 294 that no internal standard was used. Instead, we relied on matrix-matched calibration and recovery validation (spike recovery tests) to ensure quantification accuracy.

  1. Reviewer Comment 6:

Lines 94, 110, 187: A non-existent reference is indicated; it is probably meant to be two separate citations.

Response:

This has been corrected. The mistakenly combined references were separated with appropriate commas in all relevant lines (Lines 129, 251, and 267).

  1. Reviewer Comment 7:

Lines 241–242, Table 7 (Line 248): Units switch between µg/L and µg/kg. Please standardize.

Response:

We appreciate this observation. The units have now been standardized to µg/kg throughout the manuscript, and we have double-checked all conversions for accuracy.

  1. Reviewer Comment 8:

Line 251:  Please correct the typographical error ‘Espike recovery’ to ‘spiked recovery’.

Response:

Corrected as suggested. Thank you.

  1. Reviewer Comment 9:

Table 9 (Line 266): The column contains untranslated text (‘未检出’). Please replace with a clear English equivalent such as ‘Not detected (ND)’.

Response:

We have replaced the Chinese text ‘未检出’ with ‘Not detected (ND)’ in Table 6 as requested.

Round 2

Reviewer 2 Report

Comments and Suggestions for Authors There is a missing quote in line 93